# Just Play Cognitive Modern Board and Card Games, It’s Going to Be Good for Your Executive Functions: A Randomized Controlled Trial with Children at Risk of Social Exclusion

**DOI:** 10.3390/children10091492

**Published:** 2023-08-31

**Authors:** Jorge Moya-Higueras, Marina Solé-Puiggené, Nuria Vita-Barrull, Verónica Estrada-Plana, Núria Guzmán, Sara Arias, Xesca Garcia, Rosa Ayesa-Arriola, Jaume March-Llanes

**Affiliations:** 1Department of Psychology, Sociology and Social Work, Faculty of Education, Psychology and Social Work, University of Lleida, 25001 Lleida, Spain; marinasolepuiggene@gmail.com (M.S.-P.); nuria.vita@udl.cat (N.V.-B.); veronica.estrada@udl.cat (V.E.-P.); jaume.march@udl.cat (J.M.-L.); 2Centro de Investigación Biomédica en Red de Salud Mental (CIBERSAM), Instituto de Salud Carlos III, 28029 Madrid, Spain; rosa.ayesa@scsalud.es; 3Institut de Desenvolupament Social i Territorial (INDEST), 25001 Lleida, Spain; 4Atención, Familia, Infancia, Mayores (AFIM21), 04005 Almería, Spain; nuriaguzmans@gmail.com; 5Redes Cooperativa, 28025 Madrid, Spain; sara.arias@redes.coop; 6Asociación PROSEC Promotora Social, 25002 Lleida, Spain; prosec@promotorasocial.net; 7Research Group on Mental Illnesses, Valdecilla Biomedical Research Institute (IDIVAL), 39011 Santander, Spain; 8Department of Molecular Biology, School of Medicine, University of Cantabria, 39011 Santander, Spain

**Keywords:** modern board and card games, game-based intervention, flexibility, inhibition, updating

## Abstract

Modern board and card games are usually used for leisure. Few studies have focused on the type of game played in vulnerable populations. Therefore, the main aim of this study was to test the effectiveness of playing modern board and card games to enhance updating, inhibition, and flexibility in children at risk of social exclusion using games that activated specific basic executive functions. We developed a quadruple-blind randomized clinical trial during the COVID-19 pandemic. Sixty-eight participants (7–12 years old) were divided into two experimental groups: 35 children played games that directly activated basic executive functions, and 33 played games that directly triggered other cognitive domains. The primary statistical analysis consisted of mixed models. We found significant time effects in cognitive flexibility and inhibition and, to a lesser extent, in working memory in both gaming groups. We analyzed the cognitive profile of the games and found that all the games activated basic executive functions significantly, irrespective of the experimental group. Therefore, it is possible that playing any type of modern board and card game (excluding games with a high incidence of luck) could be beneficial for children at risk of social exclusion.

## 1. Introduction

According to Sousa and Bernardo [1], modern board games are different from traditional board games because the former were created and edited after 1950 by a credited author (nobody knows who was the designer of chess, but we know that Knizia designed the game Bee Alert [2]). In addition, almost all traditional board games rely on classical mechanisms, such as Grid movement and static capture [3]. In contrast, modern board games are characterized by a more innovative and larger quantity of mechanisms, such as the legacy mechanism (created in 2011) [4] and dynamics. In addition, modern board games strongly focus on aesthetics and more elaborated narratives, while traditional games are usually considered abstract games [5]. For these reasons, playing modern board and card games (also known as hobby board games) is a leisure activity with rising importance in children and adults [6].

When we play any type of board and card game, it is apparent that you are activating several cognitive processes and brain regions. Beyond subjective perceptions, when experts analyze the games, it is concluded that modern board and card games have the potential to stimulate different cognitive processes [7]. Therefore, we can speculate whether playing modern board and card games has cognitive benefits for mental health. However, we still have a few results to confirm or discard this idea.

The primary studies published about the benefits of modern board and card gaming focus on executive functions. Executive functions are high-level cognitive processes (top-down) that control lower-level processes toward direct behavior to specific goals [8,9]. They are involved in the generation, regulation, effective execution, and readjustment of goal-directed behaviors [10]. Executive functions maintain a hierarchical relationship with cognitive abilities such as attention and memory, exercising control over their functioning to adapt it to the behavioral goals of people [8,9,10]. According to the empirical model proposed by Miyake and Friedman [8,9], basic executive functions such as updating (monitoring of working memory representation), inhibition (control of dominant or automatic responses), and flexibility (shifting between mental tasks or sets) are needed to engage more complex executive functions (such as planning, reasoning, and decision making). Several studies have focused on the structure of executive functions in children [11,12,13,14,15] with mixed results. In the early stages, executive functions seem to be a unitary outcome. Then, we differentiated them into two components (updating and inhibition/flexibility). Later on in the development, they split into specific executive functions. In a systematic review and meta-analysis, Tirapu-Ostaroz et al. [11] found that these results were invariant across socioeconomic status. Therefore, having good executive function primes our biological systems and coping skills to respond well to stress. For society, the outcome is a healthier population, a more productive workforce, and reduced healthcare costs.

Social exclusion and poverty have been linked with several emotional and cognitive deficits, mainly in executive functions [16,17]. Lower socioeconomic status has detrimental consequences on brain networks [18]. Haft and Hoeft [16] proposed that these effects could be mediated by insufficient or inadequate cognitive stimulation due to economic hardships. Hence, playing board games should have cognitive benefits for children at risk of social exclusion.

According to past studies, modern board games could be an excellent playful methodology to stimulate cognitive processes and learning in children at risk of social exclusion [19,20]. Game-based interventions have been developed, usually applying games intended to activate cognitive processes without deeply analyzing the type of games used and the specific cognitive processes engaged while playing [19,20,21,22,23]. We do not know whether it is necessary to apply games that mainly activate specific basic executive functions or whether we can use games that could indirectly activate basic executive functions. Therefore, the main objective of the present study was to test the effectiveness of modern board games that mainly activate specific basic executive functions to enhance updating, flexibility, and inhibition. Specifically, we hypothesized that modern board and card games that specifically triggered basic executive functions increased updating, flexibility, and inhibition processes more than those whose primary cognitive process was not a basic executive function. Finally, we also wanted to test, in an exploratory way, whether the possible effects of the cognitive intervention based on modern board and card games were influenced by the evolution across games when playing the same game.

## 2. Materials and Methods

### 2.1. Overview

The present study was a randomized quadruple-blind clinical trial methodology. This methodology implies that neither the children themselves, the researchers, the evaluator of the results, nor the analyzer of the results knew which subjects belong to the control group and which participants belong to the experimental group [24]. In addition, this study is multi-setting research because it was executed in three social promotion entities that work with children at risk of social exclusion in different regions of Spain. The intervention lasted nine weeks (18 sessions) within their schedule of recreational activities. 

### 2.2. Participants

Before starting the investigation, the necessary sample size was calculated. A reference of 95% CI, a statistical power of 90%, and a prognosis of the durability of 15% of the sample were used. In accordance with [25], we considered 0.54 points of accuracy and 0.15 points of variance. The results indicated that 11 participants per group would be sufficient to detect significant effects [26].

Initially, 99 potential participants were contacted. The parents/guardians of the children who attended the activities for the present research, aged between eight and twelve, were informed when selecting the participants. The inclusion criteria were: (1) to study in Almeria, Madrid, or Lleida; (2) to participate in activities organized by the social promotion entities; (3) to have between 7 and 12 years old; (4) to meet the criteria of the indicator of risk of social exclusion and poverty in Spain (for instance, elevated unemployment rates, school dropouts, and deficits in housing and public spaces [27]). Exclusion criteria were: (1) the parents/legal guardian did not sign the informed consent; (2) to have language problems making it difficult follow the instructions of the study and intervention properly; (3) not having any other psychological or physical condition (i.e., intellectual developmental disorder, autistic spectrum, etc., reported by the social promotion entities). 

Figure 1 depicts the flow diagram of the study. Two children did not meet the inclusion criteria, leaving a total sample of *n* = 97. Participants were randomly allocated into an experimental group (*n* = 48) and an active control group (*n* = 49). The study was conducted during the COVID-19 pandemic so no social promotion entity granted that all the participants could follow all the sessions. We removed from the final analyses those participants who were absent in the evaluative sessions (pre or post), as it was not possible to evaluate them in any way, as well as those who has not attended at least 65% of the duration of the sessions of each of the games. We lost 29 participants for the reasons mentioned above. As shown in Figure 1, we lost an equivalent number of participants in both groups. Hence, the final sample consisted of children from 7 to 12 years (M = 9.98, Sd = 1.24, 50% female). They were attending between the third and sixth grade of primary education. We had enough participants to find significant effects based on the calculated sample size. However, we performed an attrition analysis to assess whether the retained sample showed differences from the excluded participants or not. We did not find significant differences in age (*t* (95) = −0.299, *p* = 0.766, *d* = 0.066) or in academic course (*t* (95) = −1.302, *p* = 0.196, *d* = 0.286). We even did not find differences in the distribution of sexes (χ^2^ (1) = 1.76, *p* = 0.185). 

### 2.3. Instruments

#### 2.3.1. Main Outcomes

In line with past studies [21,22,23,28], the primary outcomes of the present study were the basic executive functions, according to Miyake and Friedman [8,9]: updating (monitoring the items in the memory to replace old non-adequate material with new relevant information), inhibition (stopping an automatic response to focus on a task which needs attentional control), and flexibility (specifically, shifting between two different mental sets). As Miyake and Friedman [8,9] state, the basic executive functions are supposed to be active when we need complex cognitive processes. Therefore, strengthening these executive functions should encompass several benefits for child development. A secondary outcome was selective attention. Selective and sustained attention are important cognitive processes to perform any task but should not be part of the executive functions [29,30]. Finally, we controlled the effect of fluid reasoning, as past studies showed strong relations between executive functions and fluid reasoning [31], and this variable can influence the comprehension of the games’ rules, etc. The specific instruments to assess all the variables are explained below.

Updating was assessed with the Keep Track Task (KTT). We adapted the verbal and visuospatial KTT proposed by Tamnes et al. [32]. Both tests show a series of stimuli (words to assess verbal updating and iconic faces to assess visuospatial updating). Stimuli change during the test in a pseudorandom way. The objective was to remember the last stimuli in each trial (the last word of each semantic category in the verbal updating task and the last position of the face of each color in the visuospatial updating task). Nine trials (four with three elements, four with four elements, and one with five elements) were presented to each participant. Each test lasted 10 min. The stimuli were presented with tablets. The specific outcomes were the total hit score for verbal and visuospatial updating. The KTT is a valid and reliable measure of the updating process [33,34].

Inhibition and flexibility were assessed by the Five Digits Test (FDT) [35]: It consists of boxes with different numbers inside; according to the instructions, you must do various tasks. The first one consists of reading the number (from 1 to 5) in each cube, ignoring the number of numbers in each cube. In the second task (just a training task), respondents must count the number of asterisks inside each cube. The third task, called choice, consists of saying the number of numbers in each cube by avoiding reading the specific number. The fourth and last task is called alternation. This task presents two types of cubes: narrow and broader. Respondents are requested to say the number of numbers given in narrow boxes, avoiding the specific number presented, and the particular number in wider boxes, avoiding the amounts of numbers presented. The flexibility score was calculated by subtracting the time spent on the choice task (third) from the reading task (first). At the same time, inhibition was estimated by subtracting the alternation task (fourth) from the reading task (first). The estimated time of this paper-and-pencil test was 10 min. Hence, the outcomes analyzed in the present study were the time taken to finish the tasks, the number of errors in the first, third, and fourth task, and the flexibility and inhibition scores. Considering that flexibility is a multi-dimensional construct [36,37], the present study was focused on task switching (switching between tasks with different instructions given some stimuli [36]). The FDT showed good psychometric properties [35].

#### 2.3.2. Secondary Outcomes

Sustained attention was tested with the Face test of Perception of Differences [38] to assess attentional capacity and impulsivity control. It consists of a perceptual test composed of 60 graphic elements. Each element is formed by a set of 3 schematic drawings of faces with very elementary strokes (mouth, eyes, eyebrows, and hair). The objective is to determine which of the three faces is different and cross it out. The estimated completion time of the test is 6 min, presented in a paper-and-pencil format. The outcomes assessed were net hits (hits minus errors; A–E) and the index of impulsivity control (ICI). The Face test of Perception of Differences showed good psychometric properties [38].

Evolution along sessions was evaluated with game session record sheets: an ad hoc scale was created for the present research. After each session, participants should indicate the games they played, how many points they earned, whether they liked the game or not (on a Likert scale from 1—very bad to 10—very good), and an estimation of the whole session (on a Likert scale from 1—very bad to 5—very good). The primary outcome used was the number of points earned in each game.

#### 2.3.3. Confounding Factors

Fluid reasoning was assessed as a confounding factor with the TONI-4, nonverbal intelligence test form A, which demonstrated outstanding psychometric properties [39]. It is a test to measure intelligence quotient (I.Q.) or fluid intelligence, and it has a non-verbal format and estimated execution time of 10 min. It presents patterns of abstract figures that must be completed in ascending order of difficulty. Each element incorporates one or more of the eight salient features: shape, position, direction, rotation, contiguity, shading, size, and motion. The abstract and figurative content of these items, along with the elimination of language, reduces the cultural burden of the test. Performance is not affected by prior information or exposure. The raw score was analyzed in the present study.

### 2.4. Procedure

The methodology of the intervention program was approved by the Clinical Research Ethics Committee associated with the University of Lleida (CEIC-2614; approved 26 June 2020). The families were then contacted and the project was explained. Those who volunteered were considered participants after signing informed consent. Then, the pre-intervention evaluation phase began. In individual sessions, psychologists from each social promotion entity applied each test. The assessment time was approximately 50 min/child.

Randomization in each location was performed after assessing all the participants. The research team conducted the randomization process by listing all the codes of participants delivered by the social promotion entities and applying the randomization formula of the Excel software. The person who proceeded with the randomization process had no information about the children (just codes). The team coordinator gave the person in charge of the randomization process a blind code (1 = control group and 2 = experimental group).

Then, the intervention began and lasted nine weeks, with two sixty-minute play sessions each week. According to past research, 18 sessions should be enough to find significant differences in this type of intervention [21,23,28,40]. Participants played two games in each session, as in previous studies [19,21,23]. After ending each session, the children filled out the game session record sheet. Each child’s game records of “the game session record sheets” (See Appendix A) were entered into another database. After the intervention, the same cognitive performance tests were re-administered except for Toni 4. The psychologists of each social promotion entity who assessed the sample after the intervention were masked. Then, compensation sessions were performed by reversing the intervention to compensate for cognitive training and ensure the benefit of all participants.

We took into account three variables to select the games. The first was the primary cognitive process the game intended to activate when playing. This decision was taken by analyzing the dynamics of the games when playing with them (see Appendix A for a full description). This task was performed by two researchers according to their experience with neuropsychology and games, as in previous studies [19,21,23,28]. The experimental group played games that apparently activated the basic executive functions. The games used were Jungle Speed [41] and Ghost Blitz [42] for inhibition and flexibility, and Bee Alert [2] and Deja Vú [43] for updating. The control group played games that activated other cognitive processes than the basic executive functions. The games used were Dixit [44] to activate the Theory of mind processes, Mixmo [45] and Story Cubes [46] for verbal fluency, and Mmm! [47] for hot executive functions (decision making under risk). The other two criteria for selecting games were regarding playfulness and complexity of the games, as suggested by board game researchers [1,4]. Playfulness was estimated according to the average rating in the Board Game Geek (BGG) [48], the reference database of tabletop games. The average rating is the amount of liking assessed on a Likert scale from 1—it is so bad that you will never play that game—to 10—it is so good that you will spend all your time playing that game. The selected games are considered good games with enough fun (average rating scores between 6.26 and 7.23). We used the BGG [48] criteria to determine the game’s complexity. The complexity rating is understood as how difficult a game is to understand. All games selected could be considered easy or light games (complexity ratings between 1.07 and 1.47).

Finally, after introducing all the data into a database, the person in charge of analyzing the results proceeded. The team coordinator gave the person who conducted the analyses a blind code (1 = control group and 2 = experimental group). The person who analyzed the data did not know the correspondence between the blind code and the experimental group.

It should be noted that this was a multicenter study executed during the COVID-19 pandemic. Therefore, each center followed its course around the intervention; even so, all complied with the same temporality in the different phases of the assessment and intervention.

### 2.5. Statiscial Analyses

First, the possible differences between the two groups were calculated for the different outcomes at baseline. We executed the Mann–Whitney U tests because some variables did not follow the normal distribution (assessed by the Shapiro–Wilks test).

The effect of cognitive training was studied with the linear mixed model analysis [49]. This is a robust analysis with non-normal variables [50]. The time elapsed between the two evaluations, pre- and post-intervention, was the intraindividual factor of the study phase. The interindividual factor was the condition of cognitive training (experimental group versus active control group) to evaluate the desired effects of the intervention. Setting (each social promotion entity) was included to control its random effect. Fluid reasoning was also introduced as a confounding factor. To obtain accurate results and avoid bias, we controlled (removing them from the database) the data of the pre-evaluations of the subjects who did not complete 65% of the intervention. 

After performing the mixed models, we needed to analyze the cognitive profile of each game. We used the methodology proposed by [7].

Finally, we performed a Latent Grow Curve Modeling analysis to determine if the evolution along the different times each player played each game modulated the intervention results [51]. 

## 3. Results

### 3.1. Descriptive Analyses and Baseline Comparison

Table 1 presents the demographic characteristics and mean cognitive performance of each group. We did not find significant differences between the control and experimental groups at baseline in most of the outcomes, except in the alternation time (*U* = 401, *p* = 0.020, *d* = 0.3269). We can conclude that both groups started the study with similar cognitive skills.

We also compared the retained sample with the excluded participants in the baseline outcomes. We found only one difference between the groups (*t* (95) = −3.56, *p* < 0.001, *d* = 0.78): the excluded sample showed lower levels of linguistic updating capacity (M = 14.467, Sd = 8.460) than the retained sample (M = 19.269, Sd = 4.769). In all the other variables, the samples were equivalent.

### 3.2. Pre-Post Comparisons

The main results of the present study can be seen in Table 2. We found significant intraindividual effects (pre- and post-changes) in the outcomes assessed with the 5-Digists test. Specifically, we found that all the sample reached lower scores after the intervention in flexibility (*F* = 24.829, *p* < 0.001) and also in inhibition (*F* = 8.915, *p* = 0.004). We found that both groups increased the net hits of the Faces test (*F* = 4.880, *p* = 0.037). Finally, we found a phase-by-group interaction in the visuospatial updating (*F* = 4.756, *p* = 0.033). After the intervention, we found higher hits in the visuospatial updating task in the control group than in the experimental group. See Figure 2 to find the specific effects.

To sum up, we found that participants improved their inhibition, flexibility, and sustained attention independent of the type of game they played. In addition, the active control group was the one that showed more significant improvements compared to the experimental group.

### 3.3. Cognitive Profile of the Games

One explanation for the above-mentioned surprising results is that games could activate more cognitive processes than initially suspected. Then, we analyzed each game’s cognitive profile following Vita-Barrull et al.’s [7] suggestions. Figure 3 represents all the games used in the present study. All the games activated most of the cognitive domains initially proposed. However, the games in the active control group also triggered basic executive functions. For instance, flexibility was the main secondary cognitive process active when the active control group played games. Flexibility was the basic executive function where we found greater changes after the intervention, irrespective of the experimental group. In addition, the games played by the active control group activated a wider variety and a greater intensity of complex executive functions than the games used in the experimental group.

### 3.4. Evolution in the Execution through Plays

To test whether the evolution of scoring through the intervention sessions modulated, we first performed Latent Class Analysis in each game. Appendix A shows all the fit indexes to select the adequate model in each game. In the experimental group, only two games allowed us to discriminate between two significantly different profiles of games: Deja-vu and Ghost Blitz. In both games, one group of players slightly increased the points earned at the end of each game (increasers), and the other group slightly decreased the points earned (decreasers). We performed the same analysis with the games of the control group, and we found the same pattern of results for Mmm! and Mixmo. However, the best solution for story cubes was grouping the sample into three groups: slight increasers, slight decreasers, and without changes.

In the next step, we reanalyzed the effects of the intervention by adding the group of players as a factor to the mixed models. We wanted to know whether the impact of the cognitive intervention was different among the different groups of players according to their execution across the other times they played the games. As seen in Appendix A, we only found significant moderating effects on the group that played Ghost Blitz and Mixmo. In Ghost Blitz, the main benefits in cognitive flexibility were found for the players who slightly decreased the points earned play after play (F = 4.721, *p* = 0.036). Similarly, Mixmo’s main benefits in visuospatial working memory were found in the group of players that slightly decreased the points earned across the plays. As shown in Figure 4, the group of decreares showed lower levels of flexibility and visuospatial updating at baseline.

## 4. Discussion

The study’s main aim was to test whether the effects of a cognitive intervention based on modern board and card games were modulated by the type of game used in a sample of children at risk of social exclusion. We compared board games that were supposed to activate basic executive functions directly with board games that activated basic executive functions indirectly (or in no way). We hypothesized that playing with games that directly triggered the executive functions should entail greater improvements in those variables. However, we found that both groups improved the same amount in flexibility, inhibition, and sustained attention (See Table 2). Past research showed similar results when comparing a board game intervention with active socialization activities [52]. Surprisingly, the only different effect of the intervention was in the opposite direction from that hypothesized. The control group, those who played board and card games that were expected not to directly activate basic executive functions, was the one that showed more significant improvements. These paradigmatic results also occur in other playful disciplines. In video game research, some studies [53,54,55] have found various games that enhanced unpredicted cognitive processes and vice versa.

To better understand the unexpected results, we performed post hoc analyses by identifying the neuropsychological profile of each game (See Figure 3). First, we found that the main cognitive domain active in each game coincided with the one initially hypothesized. Second, we found that all the games used with the control group activated flexibility to some degree, though this process was not supposed to be directly active while playing. Third, the games used in the control group activated a greater quantity of complex executive functions and at a greater intensity than the games used in the experimental group. Although there are still few studies about the empirical structure of executive functions in childhood [13,15], the primary theoretical model for the structure of executive functions is still Miyake and Friedman’s proposal [8,9]. Within this model, basic executive functions are required when complex executive functions are active. Therefore, taking into account all the results about the neuropsychological profile of the games, the games used in the control group activated the basic executive functions directly and indirectly. In contrast, the games used in the experimental group activated the specific executive functions more directly. The main reason for this is that the games used in the control group activated complex cool executive functions, while the games used in the experimental group activated basic cool executive functions. The neuropsychological analysis of the games also allows us to interpret the results from a more traditional cognitive intervention perspective. For example, according to Tajik-Parvinchi et al. [56], the results of the experimental group should entail a near-transfer effect, while the effects of the control group should be better explained by a far-transfer effect. The near-transfer effects mean improving the outcomes of the processes worked on (inour experimental group), and the far-transfer effects are related to the enhancement of processes different from those directly active in the cognitive intervention. In addition, the better results in the control group than in the experimental group (taking into account that the neuropsychological profile of the games in the control group was more diverse than in the experimental group) suggest that a multicomponent intervention is better than a single-component intervention in cognitive treatments [57]. In addition, as Diamond states [58,59], executive functioning increases when the task’s difficulty is greater. It is possible that the games in the control group, as they activated more complex executive functions, were harder to master or, at least, equally challenging than those in the experimental group. Finally, another explanation for improving the performance of basic executive functions by playing can be given, considering executive functions as a unit. Together, they act as an integrated system of supervision and control and play an important role in cognitive functioning, behavior, emotional management, and social interaction; therefore, training one cognitive sub-domain affects the others [60,61]. However, some of these ideas are speculations, so more research is needed.

In addition to the effects of cognitive training (repeating a cognitive task several times to improve the performance in cognitive tasks), playing board games should positively affect cognition because of the intrinsic characteristics of playing board games. An essential element to differentiate board games from other types of games is that board games are based on rules that must be freely accepted and followed by the players [62]. Observational studies have found that players engage in different behaviors [63,64,65] that could be linked to executive functions. For example, following the rules of a board game implies that you must refrain from any behavior other than what you need to do in the game (activating cognitive inhibition). Another example is when the game ends and one player has won. If he/she does not control his/her way of celebrating the victory, the other players could walk away [63]. Or the mere fact of waiting for the turn and making all the movements to end a turn properly [63,64,65]. Therefore, playing any type of board game requires activating the executive functions internallu. We cannot discard that the beneficial effects in the control group were also partially explained by the general activation of executive functions while playing board games.

Finally, we focused on a particular characteristic when playing. It is supposed that when you have played a game several times, you can master it [62,66,67]. However, individual differences can be found with players who usually win and who typically lose. As far as we know, this is the first study to do a post hoc analysis focused on the moderating effects of winning or losing within a board game intervention. By performing Latent Class Analyses of the evolution of points earned during all the times a game was played, we differentiated two groups of players in almost all the games: players who slightly increased and players who slightly decreased the points earned (See Section 3.4 and Figure 4). When we included this characteristic in the analyses, we found that the group of slight decreasers in Ghost Blitz and Mixmo gained more significant cognitive improvements than the slight increaser group. We also found slight decreasers in both games, those with lower scores on the cognitive processes at baseline. Therefore, we found that the cognitive intervention was most beneficial to players with poorer basic executive functions at baseline. However, they were the players who slightly decreased their performance play after play. If the cognitive enhancement effect of playing board games is explained because it is like cognitive training (performing the same cognitive task repeatedly [22,23]), it makes sense that those players with better cognitive levels outperform players with lower cognitive levels. As Sala and Gobet [62] found, we can assume a significant correlation between the cognitive level of players and their performance when they play cognitive games. However, the point is that slight decreasers continued playing, though they earned slightly lower points play after play. When we play board games, different behaviors are related to psychological factors that can explain why we keep playing while losing [68,69,70]. Although most of the literature on motivation while playing comes from video games [71], it is usually thought that the same experiences are found in board games. Therefore, the primary reason for playing is how you feel after playing, reflecting a sense of “games as a therapy” [71]. In our research, players who lost were not unmotivated to play but kept playing because of the different emotional and social benefits of playing board games. As a result, they benefited most because they were performing cognitive training while they played.

### 4.1. Clinical Implications

The present results are in line with past studies on the positive effect of playing board games on children at risk of social exclusion [19]. According to our results, we can expect more significant cognitive enhancement effects in basic executive functions when children play games that activate complex executive functions (though it is not intuitive that they are triggering basic executive functions) than when they play games that activate basic executive functions (though it is apparent that they trigger basic executive functions). In addition, it seems to be an appealing procedure, especially for children with lower executive functioning, because they benefit most when they play, although they slightly decrease the points earned (with the consequence that they often lose the game). 

### 4.2. Limitations and Possible Studies

A limitation that compromised the global sample of the intervention was the pandemic situation. Due to repeated lockdowns, some children could not attend continuous play sessions, and consequently, their data were excluded from the study. 

On the other hand, having a control group that did not receive intervention with modern board and card games and another that played board games and modern cards of chance (which do not require mental effort for their execution) would have provided more data on the impact of the use of this type of game to perform cognitive training. Some studies have found that the basic executive functions keep developing through childhood [72]. Therefore, with the present study, we cannot reject that both groups improved in the cognitive domains because of evolution and not because of playing board games. However, considering past research with similar methodology [19,21,23,28,73], it is rational to interpret that the main explanation of the cognitive enhancement found in the present study, which lasted a few weeks, is due to the board and card game intervention. Future studies should address this issue.

The intervention lasted nine weeks. Past game-based methodology studies comprised an even smaller number of sessions [19,21,23] and showed significant positive results. However, standard cognitive interventions are recommended to last more sessions to find more substantial effects [74]. Therefore, in future studies, we will use more sessions.

We also suffered from a task impurity problem [75]. We only used one task per process and even one test to assess two processes (the FDT to assess flexibility and inhibition). Therefore, in future studies, it will be better to use separate tests for each executive function and at least two tests per cognitive process.

Finally, it is important to emphasize one crucial conclusion that was first shown in Vita-Barrull et al.’s study [7] and remarked in the present study. Selecting the games for a game-based intervention to activate specific cognitive processes is not easy or trivial. The best way of doing so is by analyzing the cognitive profiles of the games in the most objective way possible. However, we used Vita-Barrull et al.’s method as a post hoc analysis because the study by Vita-Barrull et al. [7] was published after we designed and began to collect the present study’s data. Ideally, we needed games that solely activated the experimental group’s specific basic executive functions and games that did not trigger any basic executive function. As shown in Figure 3, it seems that all the games used activated at least the basic executive functions (and the games in the control group activated the complex executive functions even better than the games used in the experimental group). Future studies must focus on this topic. Could it be possible that all the games that are not entirely based on luck activate at least the basic executive functions? A different selection of games could modify the results of the present study.

## 5. Conclusions

Playing modern board games seems beneficial for children at risk of social exclusion to enhance their basic executive functions. We found better results when children played games that activated complex executive functions directly and basic executive functions indirectly, in comparison with games that mainly triggered basic executive functions. In addition, playing board games seems to be especially good for children with lower baseline levels of basic executive functions, though they slightly decrease the points earned play after play. To sum up, the present study shows that playing modern board games helps children’s cognitive development when they are at risk of social exclusion. It also allows us to understand better how we choose the games to apply them as a cognitive intervention.

## Figures and Tables

**Figure 1 children-10-01492-f001:**
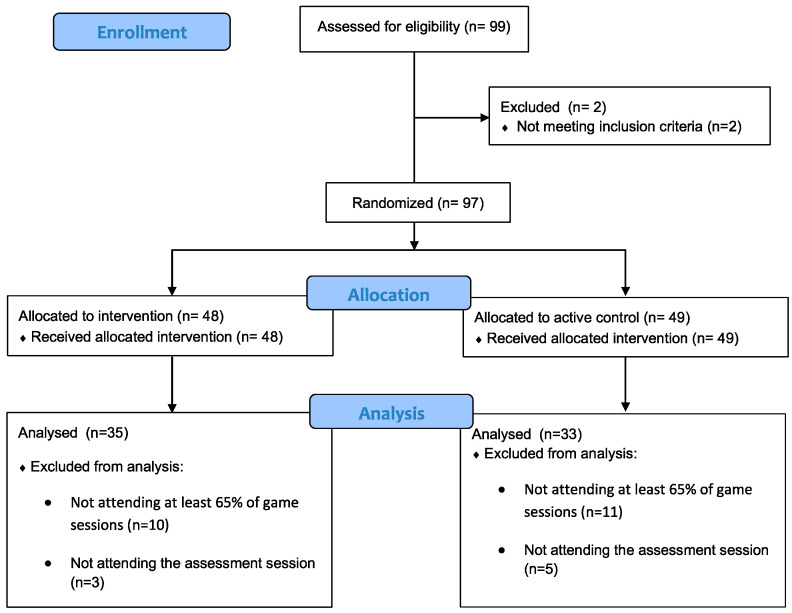
Flow diagram.

**Figure 2 children-10-01492-f002:**
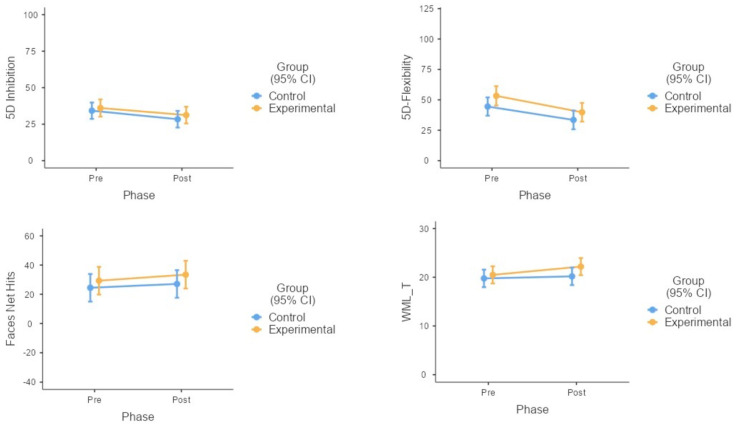
Cognitive effects of the board games intervention in both experimental groups.

**Figure 3 children-10-01492-f003:**
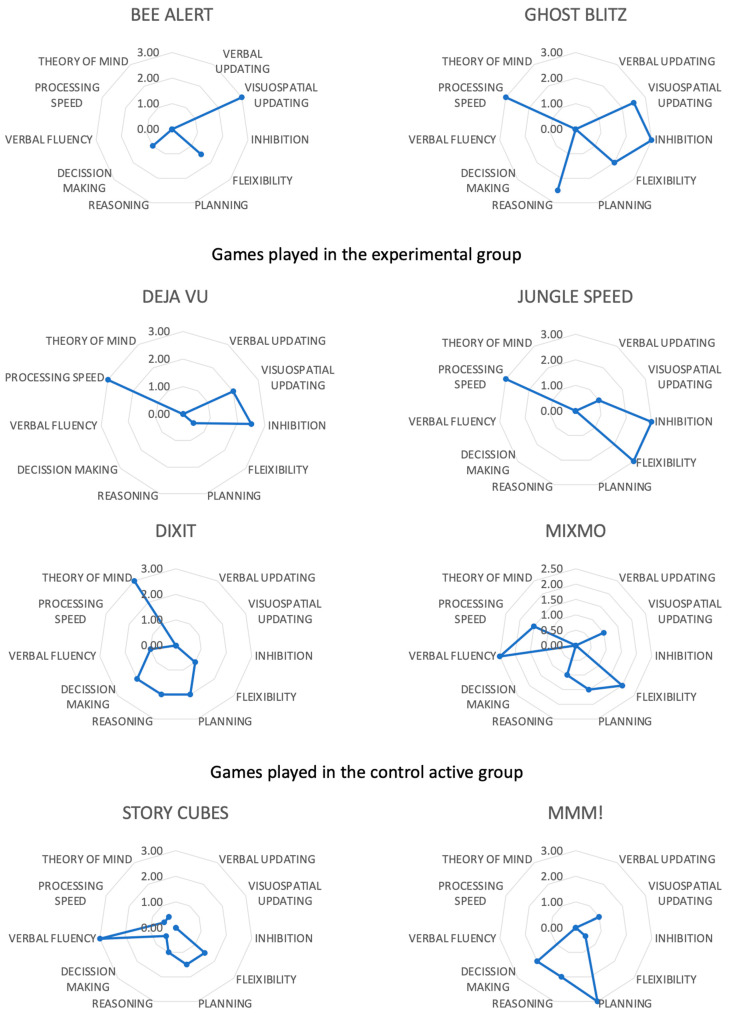
Cognitive profile of each game played.

**Figure 4 children-10-01492-f004:**
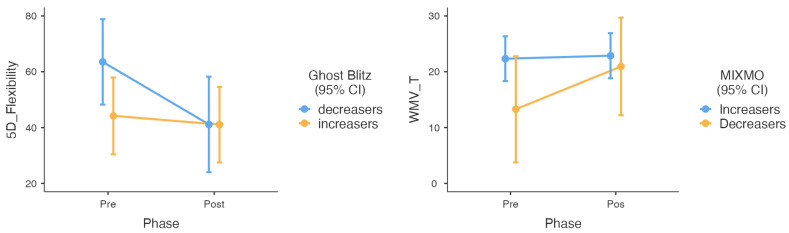
Moderation effect of the type of group according to the evolution in execution (points earned) across the plays in Ghost Blitz and Mixmo games. The outcome of the flexibility test was time, so the lower the score, the better. The updating test (WMV) outcome was hits, so the higher the score, the better.

**Table 1 children-10-01492-t001:** Descriptive analyses of each primary outcome and differences in baseline between experimental groups.

	Experimental Group (*n* = 35)		Control Group (*n* = 33)			
	Median	Interquartile Range	Median	Interquartile Range	U	d
KTT—Linguistic working memory	18	7.0	19.5	6.0	605	0.039
KTT—Visuospatial working memory	19	8.5	17.0	6.5	511	0.0875
5D—Reading time	32	9.5	30.5	8.0	533	0.1050
5D—Errors in reading	0	0	0	0	565	0.0504
5D—Time of choice	65	20.0	57.0	17.5	448	0.2471
5D—Errors in choice	2	5.5	2.0	4.0	560	0.0588
5D—Alternation time	76	29.0	66.0	20.8	401 *	0.3269
5D—Alternation errors	3	5.0	2.0	4.75	584	0.0193
5D—Inhibition	32	22.5	28.5	12.8	507	0.1479
5D—Flexibility	38	26.5	35	23	454	0.2378
FACES—Net hits	25	14	27.5	12.5	97.5	0.0051
FACES—ICI	92.6	13.8	90.8	31	88	0.1020

Note. * *p* < 0.05 U = Mann–Whitney’ U test.

**Table 2 children-10-01492-t002:** Pre-post comparison of cognitive performance between the two experimental study groups.

	Experimental Group (*n* = 35)				Control Group (*n* = 33)						
	Pre		Post		Pre		Post		F	F	F
	Mean	Sd	Mean	Sd	Mean	Sd	Mean	Sd	Phase	Group	Phase × Grupo
KTT—Linguistic working memory	20.483	0.889	22.180	0.889	19.76	0.905	20.180	0.905	**2.920** ᵻ	1.61	1.070
KTT—Visuospatial working memory	19.533	1.026	19.225	1.026	18.473	1.049	20.978	1.049	**2.901** ᵻ	0.074	**4.756** *
5D—Reading time	34.717	1.691	31.613	1.624	33.414	1.588	30.814	1.627	**5.340** *	0.312	0.042
5D—Errors in reading	0.157	0.090	0.137	0.087	0.191	0.085	0.268	0.087	0.223	0.644	0.633
5D—Time of choice	70.900	3.596	62.832	3.512	67.469	3.436	59.229	3.485	**20.085** ***	0.596	0.009
5D—Errors in choice	0.751	0.179	0.500	0.171	0.505	0.167	0.180	0.172	**4.274** *	2.160	0.070
5D—Alternation time	88.009	4.370	71.402	4.266	77.914	4.174	64.213	4.235	**44.806** ***	2.577	0.412
5D—Alternation errors	3.752	0.736	2.415	0.711	3.336	0.695	1.770	0.710	**9.183** *	0.385	0.058
5D—Inhibition	36.072	2.968	31.216	2.878	34.232	2.816	28.398	2.869	**8.915** **	0.435	0.075
5D—Flexibility	53.285	4.000	39.784	3.874	44.495	3.790	33.469	3.863	**24.829** ***	2.548	0.253
FACES—Net hits	29.298	4.613	33.423	4.613	24.439	4.596	27.064	4.596	**4.880** *	0.785	0.241
FACES—ICI	88.748	8.789	86.282	8.789	72.117	8.757	69.958	8.757	0.496	1.895	0.002

Note. ᵻ *p* < 0.10 * *p* < 0.05 ** *p* < 0.01 *** *p* < 0.001. Bold: significant results. KTT = Keep Track Task. 5D = 5 Digits Test. FACES = Face test or Perception of Differences. ICI = Index of Cognitive Impulsivity.

## Data Availability

Data are available on request.

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
