# Peer review of "Just Play Cognitive Modern Board and Card Games, It’s Going to Be Good for Your Executive Functions: A Randomized Controlled Trial with Children at Risk of Social Exclusion"

_children, 2023, doi:10.3390/children10091492_

Round 1
Reviewer 1 Report
This interesting paper examines the effectiveness of playing modern board and card games to enhance basic executive functioning in children at risk of social exclusion employing games that triggered specific EF. The topic becomes significant and interesting to the journal’s readership. Though I consider that this work is highly important and interesting for the neuropsychological field and researchers, I do have some suggestions to strengthen the manuscript.
- Considering that the structure of EFs changes throughout development, when defining the components of EFs it is suggested to also refer to studies that have examined the EF structure in children. To date, there are studies that have examined the structure of EF in Spanish-speaking children and that have also examined their invariance across socioeconomic status (SES).
- Regarding the tasks employed to assess EF, it is not the best practice to use the same test to assess two EFs, due to the task-impurity problem. While the five-digit test assesses interference and is based on the Stroop paradigm, it mainly assesses the inhibition component. The fact that the test also demands other EFs such as cognitive flexibility does not mean that it appropriately assesses this component. To date, there are various instruments with adequate psychometric properties to assess both reactive (e.g., TMT, WCST) and spontaneous (e.g., verbal fluency tasks) cognitive flexibility. Also considering that cognitive flexibility is a multidimensional construct, it is suggested to specify which specific component was evaluated.
- It is recommended to add more information as regards the psychometrical properties of the cognitive tasks employed, namely, for example, data on its validity and reliability.
- The fact that the same task was used to assess inhibition and flexibility should be included as an important limitation of the study.
Author Response
Response to Reviewer 1 Comments
C1. This interesting paper examines the effectiveness of playing modern board and card games to enhance basic executive functioning in children at risk of social exclusion employing games that triggered specific EF. The topic becomes significant and interesting to the journal’s readership. Though I consider that this work is highly important and interesting for the neuropsychological field and researchers, I do have some suggestions to strengthen the manuscript.
Response: Thank you for your kind comments.
C2. Considering that the structure of EFs changes throughout development, when defining the components of EFs it is suggested to also refer to studies that have examined the EF structure in children. To date, there are studies that have examined the structure of EF in Spanish-speaking children and that have also examined their invariance across socioeconomic status (SES).
Response: Thank you for your suggestion. It is true that we did not mention anything about EF structure or development. We have added some sentences about it. See L73-79.
C3. Regarding the tasks employed to assess EF, it is not the best practice to use the same test to assess two EFs, due to the task-impurity problem. While the five-digit test assesses interference and is based on the Stroop paradigm, it mainly assesses the inhibition component. The fact that the test also demands other EFs such as cognitive flexibility does not mean that it appropriately assesses this component. To date, there are various instruments with adequate psychometric properties to assess both reactive (e.g., TMT, WCST) and spontaneous (e.g., verbal fluency tasks) cognitive flexibility. Also considering that cognitive flexibility is a multidimensional construct, it is suggested to specify which specific component was evaluated.
Response: It is a very wise comment, we must say. It is true that the five-digit test has its limitations. The main reason of using it was that is relatively short to administrate. The project was developed during the COVID-19 pandemic, and we needed to balance tests between good psychometric properties and fast to use. We have added one sentence when we describe the instrument (See L170-220).
C4. It is recommended to add more information as regards the psychometrical properties of the cognitive tasks employed, namely, for example, data on its validity and reliability.
Response: We have added some sentences about the psychometric properties of the tests. See See L170-220.
C5. The fact that the same task was used to assess inhibition and flexibility should be included as an important limitation of the study.
Response: Thanks for the comment. The reviewer is right. We have added some sentences aboutthis limitation. See L690-693.

Reviewer 2 Report
Dear Authors.
Very interesting theme. Congratulations.
In my opinion, the article has some weaknesses: some simple and easily overcome, others more profound, which need to be reflected upon and possibly changed.
I present my reflections according to your article.
Title
-“Just play to any kind of modern board and card games, it's going to be good for your executive functions: a randomized controlled trial with children at risk of social exclusion”- The title is appealing, however, I think it may bias the reader because there are many games that are about luck and not promoting reasoning - the authors should think about this reformulation.
Abstract
- Lines 26 e 27- "basic executive functioning in children"- In my opinion, since you don't assess FE in its generality you should identify the basic dimensions you assessed and not generalize (cognitive flexibility, inhibition, working memory).
- “Keywords: Modern board and card games, cognitive intervention, flexibility, inhibition, working memory”- There is no direct relationship between these and the summary - here you talk about the FE that you evaluated. I also don't think that one of the key words should be cognitive intervention because you didn't do that, what you did, from my perspective, was to promote the facilitation of guided play sessions.
Introduction
- Lines 41-44- “modern board games are different to traditional board games because the formers have been created and edited after 1950, with innovative and greater quantity of mechanisms [2] and dynamics than traditional board and card games.”- I think you should explain these concepts better - what are modern board games really? What makes them different from traditional ones? How were these differences validated?
2. Materials and Methods
2.1 Overview
- Line 91- Infants seems to be a terminology applied to younger children, in the sample the children are between 7 to 12 years old, it would be, in my view, to review the terminology.
- “The intervention lasted 6 weeks, within their schedule of recreational activities.”- Why 6 weeks? Where do you base this reduced time? It would be important to explain. Not least because the children, like all of us, were in an atypical period of the COVID-19 pandemic, which in itself may bias the results. This is a reality that the literature has been reporting. These issues, in my view, should be explained here or, if you can't support them, asked in the limitations of the study.
2.1. Participants
- Lines 103-105 – “To make the selection of the participants, the parents/guardians of the children who attend the associations Afim21, Prosec and Redes…”- For ethical reasons, I don't think you should identify the institutions but rather describe them briefly (mission/objectives).
- Line 108- “4) To meet the criteria of the indicator of risk of social exclusion and poverty in Spain”- It would be interesting to make these criteria known.
- Lines 110-111- “2) To have language problems to follow the instructions of the study and intervention properly.”- "- What about Intellectual Developmental Disorder issues? Autistic Spectrum? Psychotropic drug use? Neurological problems? Psychiatric problems? The literature reports that all of these issues directly influence the dynamics of executive functioning. Have you considered these issues? I think that if you have considered it, you should mention it. If not, this question should be asked in the limitations of the study.
- Lines 116-118- “For the final analyses we retained only those participants who were absent in the evaluative sessions (pre or post) and was not possible to evaluate them in any way…”- Do you mention that these children were removed? That is, they were not considered in the statistical analysis of your study. This statement is not clear. Although if you explain it further, I think you should clarify this paragraph.
- Lines 118-119- “… as well as the fact of not having attended at least 65% of the time to the sessions of each of the games.”- That is, did you evaluate all the children who attended the activities between 4 (3.9) to 6 weeks? It seems to me that the time is indeed too limited to talk about changes in the respective EF dimensions. In my view, you should support this with previous studies that indicate that 2h in 6 weeks (12 hours of stimulation) can be significant. It also seems frankly little to me that some of the children did not attend all of the sessions, although this information was not made known. Which I rate as another limitation.
- You don't describe the participants- age, gender, educational background, number of sessions that the children attended... in my view it would be fundamental to have access to this information.
2.3. Instruments
2.3.1. Main outcomes
Lines 127-128- “The main outcomes of the present study were the basic executive functions according to Miyake & Friedman[5,6]: updating, inhibition and flexibility”. I think you could be more strict in the terminology you adopt. This is because, according to, doi:10.1006/COGP.1999.0734. “This individual differences study examined the separability of three often postulated executive functions-mental set shifting ("Shifting"), information updating and monitoring ("Updating"), and inhibition of prepotent responses ("Inhibition")” and according to doi: 10.1016/j.cortex.2016.04.023, “They include abilities such as response inhibition, interference control, working memory updating, and set shifting.”
- And why the designation of basic executive functions? It would be interesting if you could explain why.
- Lines 128-130 “A secondary outcome was selective attention. Finally, we controlled the effect of fluid reasoning.”. This is the first time this information appears and you do not present any theoretical support for these dimensions. In my opinion, it would be important to review.
- Lines 153-157- “By combining the time spent in some of these tasks, we obtained a flexibility score and cognitive inhibition (resistance to interference) score. The estimated time of this paper-and-pencil test was 10 minutes. Hence, the outcomes analysed in the present study were the time to end and amount of errors in the first, the third and the fourth task, and the flexibility and inhibition scores.”- It is not clear what choice of analysis you make. It is also unsupported. In my view, you should be more specific and clearly support your criteria.
2.3.4. Sociedomgraphics
- Cricket in the title - Sociedomgraphics - should be reviewed
- Lines 183-185- “The social promotion entities provided basic sociodemographic information such as age, sex and education level of each participant.”- This point does not seem necessary here. I think this information should be included in point 2.1 (Participants) and be presented (as I mentioned before).
2.4. Procedure
- Line 195- “social promtion entities”- typo promotion
- Line 196- “fornula of the Excel software.”- typo formula
- Lines 200-201- “Then, the intervention begun and lasted six weeks, with two sixty-minute play sessions each week”- I again appeal to you to think about this- would it be important to sustain this time- 2h/week for 6 weeks (12 hours in total) and it is not clear whose time it was all- is it enough to explain the changes? I think you should think about and support this methodological decision in the literature very well.
- Lines 209-213- “We took into account three variables to select the games. The first one was the main cognitive process that the game is intended to activate when playing. This decision was taken by a subjective criteria by two researchers according to their experience on neuropsychology and on games. The experimental group played with games that, apparently, activated the basic executive functions.”- What subjective criteria? You should be clearer in this explanation. Not least because, in my opinion, this choice may have been responsible for your results. If it is not very well supported, the whole intervention can easily be equivocated.
- In my opinion, the procedure is not clear or sustained, it leaves many questions unanswered. You should be more objective, always supporting in literature the choices you have made.
3. Results
3.1. Descritptive analyses and baseline comparision
- Crickets in the title - Descritptive … comparision- should be reviewed
- Line 261- “ (U = 261 401 p = 0.020 d = 0.3269)”- missing formatting
- Lines 268-270- “(F =24.829, P<0.001)”- p should be lowercase- must review; lack of commas, sometimes you put commas, sometimes periods (,/.).
. Line 268 “ion. (F = 8.915, P” - review
4. Discussion
- “The main reason is that the games used in the control group activated complex cool executive functions, while the games used in the experimental group activated basic cool executive functions”- This conclusion is made peremptorily. Where do you support it? It would be important for you to explain it better.
- In my opinion, you make several inferences that are not supported. It should be reviewed.
CLINICAL IMPLICATIONS
- “board games seem to be more beneficial to executive functioning development than other type of games, such as videogames. “ - I don't see the relevance of this statement to the clinical implications of your study. You have not studied them, so it is not your conclusion as you state in the next paragraph.
- “In addition, it seems to be an appealing procedure specially for children with lower executive functioning, because they benefit most when they play, although the slightly decrease the points earned (with the consequence that they often lose the game).”- You again make unsupported claims and present them as fact. You should, in my opinion, review.
LIMITATIONS AND POSSIBLE STUDIES
- “However, considering past research[51,52,54,55], it is rational to interpret that the main explanation of the evolution found in the present study that lasted a few weeks, is due to the board and card games intervention.”- What do you mean? I don't understand. You didn't make this comparison in your study. Why talk about this?
- In my opinion, necessarily this point has to be developed, there are many more limitations that are not identified.
- A question for reflection: Shouldn't the games chosen be equated?
- It is not clear to me “Thus, a future study to be proposed would be to include as independent variables cognitive training with modern board and card games that directly involve basic executive functions, training with board games that do not require much mental effort, cognitive training through paper and pencil tasks, and children without cognitive training.” It should be better explained.
- I ponder: Shouldn't a more detailed evaluation of games in terms of stimulation of executive functions be proposed?
- I ponder: Shouldn't the stimulation time be a variable to be equated as well?
- I ponder: Shouldn't the fact that the children did not attend all the questions be mentioned? (Are there differences between children who attended all the sessions and those who only attended some? Even with this question we can't answer because we don't have that information)
References
Review some incomplete references, such as:
Thurstone, L.L.; Yela, M. CARAS-R. 2012.
Brown, L.; Sherbenou, R.J.; Johnsen, S.K.; Berna, G.·; Viena, ·; Oxford, ·; Boston, P.; Á msterdam, ·; Praga, ·; Copenhague, F.; et al. Grupo Editorial Hogrefe TONI-4. 2010
Vuarchex, T.; Yakovenko, P. Jungle Speed 1997.
Roubira, J.-L. Dixit; Libellud & Asmodee, 2008;
Faubet, J.; Hatesse, S. Mixmo; Asmodee, Sly Frog Games, 2009;
- It is important to review all references in detail.
Good job.
I wish you all the best.
Author Response
Response to Reviewer 2 Comments
Dear Authors.
Very interesting theme. Congratulations.
In my opinion, the article has some weaknesses: some simple and easily overcome, others more profound, which need to be reflected upon and possibly changed.
I present my reflections according to your article.
Response: Thank you for your comments. They have been of great help to improve the manuscript.
Title
C1 -“Just play to any kind of modern board and card games, it's going to be good for your executive functions: a randomized controlled trial with children at risk of social exclusion”- The title is appealing, however, I think it may bias the reader because there are many games that are about luck and not promoting reasoning - the authors should think about this reformulation.
Response: Thank your idea. Most of modern board and card games rely on planning and cognition, but it is true that some are about luck. So, we have decided to change the title to " Just play cognitive modern board and card games, it's going to be good for your executive functions: a randomized controlled trial with children at risk of social exclusion". In fact, it is shorter now. So, thank you so much another time.
Abstract
C2 - Lines 26 e 27- "basic executive functioning in children"- In my opinion, since you don't assess FE in its generality you should identify the basic dimensions you assessed and not generalize (cognitive flexibility, inhibition, working memory).
Response: Thanks. It has been changed.
C3 - “Keywords: Modern board and card games, cognitive intervention, flexibility, inhibition, working memory”- There is no direct relationship between these and the summary - here you talk about the FE that you evaluated. I also don't think that one of the key words should be cognitive intervention because you didn't do that, what you did, from my perspective, was to promote the facilitation of guided play sessions.
Response: As we have changed the abstrat to fit to the previous comment, we feel that the keywords are much better now. However, we have changed the term "working memory" for "updating" to be more precise. In addition, we believe that "game-based intervention" is a much better term of the type of intervention that we did. So, we have changed it too. See L38-39.
Introduction
C4 - Lines 41-44- “modern board games are different to traditional board games because the formers have been created and edited after 1950, with innovative and greater quantity of mechanisms [2] and dynamics than traditional board and card games.”- I think you should explain these concepts better - what are modern board games really? What makes them different from traditional ones? How were these differences validated?
Response: Thanks for the comment, but it is difficult to answer without being too technical. Anyway, we have added some sentences and examples to clarify this point. See L43-54.
- Materials and Methods
2.1 Overview
C5 - Line 91- Infants seems to be a terminology applied to younger children, in the sample the children are between 7 to 12 years old, it would be, in my view, to review the terminology.
Response: Thanks for the comment. We have changed the term "infant" by children in the two sentences we have used it. See L115 & L150.
C6 - “The intervention lasted 6 weeks, within their schedule of recreational activities.”- Why 6 weeks? Where do you base this reduced time? It would be important to explain. Not least because the children, like all of us, were in an atypical period of the COVID-19 pandemic, which in itself may bias the results. This is a reality that the literature has been reporting. These issues, in my view, should be explained here or, if you can't support them, asked in the limitations of the study.
Response: As far as we know, it is not so important how many weeks do last the intervention, but the number of sessions developed. As it is explained in P6 L227, we assured 12 gaming sessions. According to Klingberg (2010), a minium number of 10 sessions are needed to begin finding effects in cognitive training. In previous studies with specific populations, only 5 sessions were enough to find significant increases in group of children who played modern board games (Estrada-Plana, Esquerda, Mangues, March-Llanes, & Moya-Higueras, 2019a). Benzing et al., (2019) used the same temporality, showing significant results. In fact, with older adults, a game-based intervention within 6 weeks have found significant results (Estrada-Plana et al., 2021). In addition to this evidence, we thought that 6 weeks were easier to follow during the COVID-19 pandemic than a longer intervention. With more weeks to play, the more likely to have individual lockdowns and so on. Any way, we have added a sentence to justify this decission. See P6, L363-364. We have also added a paragraph in the limitations section (See L685-689).
2.1. Participants
C7 - Lines 103-105 – “To make the selection of the participants, the parents/guardians of the children who attend the associations Afim21, Prosec and Redes...”- For ethical reasons, I don't think you should identify the institutions but rather describe them briefly (mission/objectives).
Response: We have changed this sentence. See L129.
C8 - Line 108- “4) To meet the criteria of the indicator of risk of social exclusion and poverty in Spain”- It would be interesting to make these criteria known.
Response: We have added the requested information. See L133-134
C9 - Lines 110-111- “2) To have language problems to follow the instructions of the study and intervention properly.”- "- What about Intellectual Developmental Disorder issues? Autistic Spectrum? Psychotropic drug use? Neurological problems? Psychiatric problems? The literature reports that all of these issues directly influence the dynamics of executive functioning. Have you considered these issues? I think that if you have considered it, you should mention it. If not, this question should be asked in the limitations of the study.
Response: Wise comment, but we did not took took it into account because the associations assured us that all the sample have no more problems than the condicion of being at risk of social exclusion. It implies that the associations used this exclusion criteria indeed. So, we have added this information (See P3, L137-139).
C10 - Lines 116-118- “For the final analyses we retained only those participants who were absent in the evaluative sessions (pre or post) and was not possible to evaluate them in any way...”- Do you mention that these children were removed? That is, they were not considered in the statistical analysis of your study. This statement is not clear. Although if you explain it further, I think you should clarify this paragraph.
Response: The reviewer is totally right. There is a mistake in the sentence. It is puzzling. We removed from the analyses those participants who were absent in any evaluative session. We have solved this sentence (L144).
C11 - Lines 118-119- “... as well as the fact of not having attended at least 65% of the time to the sessions of each of the games.”- That is, did you evaluate all the children who attended the activities between 4 (3.9) to 6 weeks? It seems to me that the time is indeed too limited to talk about changes in the respective EF dimensions. In my view, you should support this with previous studies that indicate that 2h in 6 weeks (12 hours of stimulation) can be significant. It also seems frankly little to me that some of the children did not attend all of the sessions, although this information was not made known. Which I rate as another limitation.
Response: Please, we kindly request Reviewer 2 to see our response to C6. However, we agree that a greater quantity of sessions should be advisible. So we have included it in the limitations section (see L666-669).
C13 - You don't describe the participants- age, gender, educational background, number of sessions that the children attended... in my view it would be fundamental to have access to this information.
Response: As the manuscript is long, we decided to sum up all the information requested by Reviewer 2. It can be found in P3 L149-159.
2.3. Instruments
2.3.1. Main outcomes
C14 Lines 127-128- “The main outcomes of the present study were the basic executive functions according to Miyake & Friedman[5,6]: updating, inhibition and flexibility”. I think you could be more strict in the terminology you adopt. This is because, according to, doi:10.1006/COGP.1999.0734. “This individual differences study examined the separability of three often postulated executive functions-mental set shifting ("Shifting"), information updating and monitoring ("Updating"), and inhibition of prepotent responses ("Inhibition")” and according to doi: 10.1016/j.cortex.2016.04.023, “They include abilities such as response inhibition, interference control, working memory updating, and set shifting.”
Response: We understand the Reviewer, but we decided not to develop this topic deeper for not being redundant. We have explained what the basic executive functions are in L63-82. That is why we only use the specific terminology "the basic executive functions according to Miyake & Friedman[5,6]: updating, inhibition and flexibility" here. In addition, it is usual to find the term "flexibility" to make reference to a cluster of processes such as set-shifting, task-swithing, etc. That is why we use this general term instead of more specific words. So, considering all the changes that we have done in this review, we are not going to modify this section. However, if Reviewer 2 consider that it is a very important issue, we are open to change it.
C15 - And why the designation of basic executive functions? It would be interesting if you could explain why.
Response: As it is stated at L69-73, "According to the empirical model proposed by Miyake and Friedman (Friedman & Miyake, 2017; Miyake et al., 2000), basic executive functions such as updating (monitoring of working memory representation), inhibition (control of dominant or automatic responses) and flexibility (shifting between mental tasks or sets) are needed to engage more complex executive functions". As it is explained in the introduction section, in order to avoid being redundant, we don't explain it again in the in the 2.3.1 Main outcomes section. However, as in Comment 14, if Reviewer 2 considers that it is important to explain it in a deeper way, we are open to do it.
C16 - Lines 128-130 “A secondary outcome was selective attention. Finally, we controlled the effect of fluid reasoning.”. This is the first time this information appears and you do not present any theoretical support for these dimensions. In my opinion, it would be important to review.
Response: Thanks for the comment. It is true that we don't estate why these variables are so important to measure and analyse them. First, specific attentional processes are considered variables out of the executive functions (Loher & Roebers, 2013; Spruijt, Dekker, Ziermans, & Swaab, 2018). As the main outcomes were the basic executive functions, we treated sustained attention as a secondary outcome. Second, fluid reasoning has strong associations with fluid intelligence (Dehn, 2017). Thus, with the intention to test whether the possible cognitive gains with the game-based intervention was specific to the executive funcitons, we statistically controlled the effect of fluid intelligence. We have added a sentence justifying it (See L166-168).
C17 - Lines 153-157- “By combining the time spent in some of these tasks, we obtained a flexibility score and cognitive inhibition (resistance to interference) score. The estimated time of this paper- and-pencil test was 10 minutes. Hence, the outcomes analysed in the present study were the time to end and amount of errors in the first, the third and the fourth task, and the flexibility and inhibition scores.”- It is not clear what choice of analysis you make. It is also unsupported. In my view, you should be more specific and clearly support your criteria.
Response: Thanks for the comment. We tried to be concise in the instrument's explanation. However, it is true that a non-expert reader on the 5 digists test will not imagine how the outcomes were calculated. We have changed the explanation. See L198-201.
2.3.4. Sociedomgraphics
C18 - Cricket in the title - Sociedomgraphics - should be reviewed
Response: Thanks for noticing this mistake. Changed.
C19 - Lines 183-185- “The social promotion entities provided basic sociodemographic information such as age, sex and education level of each participant.”- This point does not seem necessary here. I think this information should be included in point 2.1 (Participants) and be presented (as I mentioned before).
Response: It is true that this sentence do not contribute with any information. We have removed it. Please, see response to Comment 13.
2.4. Procedure
C20 - Line 195- “social promtion entities”- typo promotion
Response: Thanks for noticing this mistake. Changed.
C21- Line 196- “fornula of the Excel software.”- typo formula
Response: Thanks for noticing this mistake. Changed.
C22- Lines 200-201- “Then, the intervention begun and lasted six weeks, with two sixty-minute play sessions each week”- I again appeal to you to think about this- would it be important to sustain this time- 2h/week for 6 weeks (12 hours in total) and it is not clear whose time it was all- is it enough to explain the changes? I think you should think about and support this methodological decision in the literature very well.
Response: Please, see response to Comment 6.
C23- Lines 209-213- “We took into account three variables to select the games. The first one was the main cognitive process that the game is intended to activate when playing. This decision was taken by a subjective criteria by two researchers according to their experience on neuropsychology and on games. The experimental group played with games that, apparently, activated the basic executive functions.”- What subjective criteria? You should be clearer in this explanation. Not least because, in my opinion, this choice may have been responsible for your results. If it is not very well supported, the whole intervention can easily be equivocated.
Response: As far as we know, what Reviewer 2 proposes in this comment is one of the main questions of using games in cognitive interventions. How could we be sure that the game we use in an intervention really activates specific cognitive processess before using it? Research with casual/commercialized video games have faced the same problem (Baniqued et al., 2013, 2014; Kranz, Baniqued, Voss, Lee, & Kramer, 2017). In fact, we used the same methodology than Baniqued et al. (2013). First, the analyses of what cognitive processes should be activated by each game was done by single researchers. Then, after different experimental procedures, we can be surer that the game activates specific cognitive processes. The problem is that the research in board games is still beginning. As far as we know, Vita-Barrull et al. (2022) work have been the first one to propose a methodology to being analyzing the cognitive processes when we play analog games in a more objective way. However, we designed and executed the present study before Vita-Barrull et al. (2022) study. That is why we use the Vita-Barrull et al. (2022) methodology as an exploratory analysis after the main results of our study. So, we proceded as all the previous board game studies (Benzing et al., 2019; Estrada-Plana et al., 2021; Estrada-Plana, Esquerda, Mangues, March-Llanes, & Moya-Higueras, 2019b; Vita-Barrull, Guzmán, et al., 2022), by using board games that were suposed to activate the basic executive functions. But, as Reviewer 2 explains, as we did not have the proper methodology (Vita-Barrull et al., 2022) to make the initial decission right, we found that the games of the control group indeed activated the basic executive functions after the 3.3. Cognitive profile of the games analysis. So, the present comment is really the key question of the present study. That is why we performed the 3.3. Cognitive profile of the games analysis and that is why we use several sentences of the discussion section to explain it (See L569-607).
C24- In my opinion, the procedure is not clear or sustained, it leaves many questions unanswered. You should be more objective, always supporting in literature the choices you have made.
Response: We sincerely understand Reviewer 2. We expect to having solved all these issues. If not, we are open to do it once again.
- Results
3.1. Descritptive analyses and baseline comparision
C25- Crickets in the title - Descritptive ... comparision- should be reviewed
Response: Thanks for noticing this mistake. Changed.
C26- Line 261- “ (U = 261 401 p = 0.020 d = 0.3269)”- missing formatting
Response: Thanks for noticing this mistake. Changed.
C27- Lines 268-270- “(F =24.829, P<0.001)”- p should be lowercase- must review; lack of commas, sometimes you put commas, sometimes periods (,/.).
. Line 268 “ion. (F = 8.915, P” - review
Response: Thanks for noticing all these mistakes. Changed.
- Discussion
C28- “The main reason is that the games used in the control group activated complex cool executive functions, while the games used in the experimental group activated basic cool executive functions”- This conclusion is made peremptorily. Where do you support it? It would be important for you to explain it better.
Response: Thanks for remaring one of the most important questions of the study. As we state in the previous sentences to this one, we support this idea in 3.3 Cognitive profile of the games analysis. As can be seen at Figure 3, only one game used in the experimental group activated complex cool executive functions (Ghost Blitz activating nearly completely the cognitive process of reasoning. On the contrary, in the control group, all the games activated (sometimes with more strenght and sometimes with less intensity) the complex executive functions. In fact, the difference is that the games used in the experimental group solely activate basic executive functions (with the exception of Ghost Blitz and Bee alert a bit) and the games used in the control group activated complex and basic executive functions in a more diverse way. It is possible that we did not explicited with executive functions are considered basic or complex. Thanks to Reviewer 1, we have increased the amount of information about the distinction between basic and complex executive functions. So, please, see L73-79.
C29- In my opinion, you make several inferences that are not supported. It should be reviewed.
Response: Thanks for the comment. The first paragraf (L552-568) in the discussion section is based on the main analysis of the study that can be seen at Table 2. As we have tried to explain in the results and discussion sections, to better interpret the results, we performed extra post-hoc analyses. The first consisted on the analysis of the cognitive profile of the games using the same methodology of Vita-Barrull, March-Llanes, et al., (2022). Then, the second paragraf (L569-607) of the discussion section is based on this analysis, which can be seen at Figure 3. However, we understand that there is not only one explanation to the main results, and we ponder over other ways of interpreting the results. See paragraph 3 of the discussion section (L608-623). Then, we interpret the results of the last post-hoc analysis we performed, the effect of the evolution in the execution through plays (see L624-653). So, we understand that all the assertions are based on our own analyses. We have tried to compared our results to past research citing the studies accordingly. In addition, we have tried to give theoretical explanations to the results, citing the studies properly too. So, we feel that we have argued all the inferences we have done. But we are open to discuss specific ideas.
CLINICAL IMPLICATIONS
C30- “board games seem to be more beneficial to executive functioning development than other type of games, such as videogames. “ - I don't see the relevance of this statement to the clinical implications of your study. You have not studied them, so it is not your conclusion as you state in the next paragraph.
Response: We can agree with Reviewer 2 that the assertion is too strong based only in our results (where we do not compare videogames with board games). We integrated our research with the study of Gashaj, Dapp, Trninic, & Roebers, (2021). They found that board games exerted more beneficial effects for cognitive development than videogames. But we understand that they studied a different population. So, we have decided to remove the present sentece (See L655-664).
C31- “In addition, it seems to be an appealing procedure specially for children with lower executive functioning, because they benefit most when they play, although the slightly decrease the points earned (with the consequence that they often lose the game).”- You again make unsupported claims and present them as fact. You should, in my opinion, review.
Response: The present assertion is based on the result explained at the 3.4. Evolution in the execution through plays section. Our results indicated that the children who benefited the most playing games were those who frequently lost when they played to Ghost Blitz or Mixmo. Educators from the assocations who applied the games told us that all the children were playing all the time, regardlees of whether they won or not. So, all the children who usually lost kept on playing. So, it seems that the benefits of playing probably come from the fact of playing. We understand that the assertion is fully based on our own analyses. However, considering all the comments of Reviewer 2, we could specify in the discussion section which analysis we used to argue each idea.
LIMITATIONS AND POSSIBLE STUDIES
C32- “However, considering past research[51,52,54,55], it is rational to interpret that the main explanation of the evolution found in the present study that lasted a few weeks, is due to the board and card games intervention.”- What do you mean? I don't understand. You didn't make this comparison in your study. Why talk about this?
Response: We agree with Reviewer 2 that we did not bake this comparision with our own data. But just it is why we do this assertion. The logic is the following. We had not got any passive control group. So, the increases in executive funcions found in the experimental and in the control group could be alternatively explained by maturative effects. However, considering all the past studies that we have found with the same methodolgy and with passive control groups, the best interpretation for our results are not madurative effects but the effects of the board game intervention. But, as we did not use a passive control group, we cannot be sure of it. That is why it is a limitation.
C33- In my opinion, necessarily this point has to be developed, there are many more limitations that are not identified.
Response: Thanks for the comment. We have developed this section considering the comments of all the Reviewers. See L694-718.
C34- A question for reflection: Shouldn't the games chosen be equated?
Response: Thanks for the comment. We have added it as a limiation. See L694-718.
C35- It is not clear to me “Thus, a future study to be proposed would be to include as independent variables cognitive training with modern board and card games that directly involve basic executive functions, training with board games that do not require much mental effort, cognitive training through paper and pencil tasks, and children without cognitive training.” It should be better explained.
Response: Thanks for the comment. The sentence make reference to an ideal design where we try to reject the maturative effect explanation.
C36- I ponder: Shouldn't a more detailed evaluation of games in terms of stimulation of executive functions be proposed?
Response: Thanks for the comment. We totally agree. We have added it as a limiation. See L694-71..
C37- I ponder: Shouldn't the stimulation time be a variable to be equated as well?
Response: Thanks for the comment. We totally agree. We have added it as a limiation. See L685-689.
C38- I ponder: Shouldn't the fact that the children did not attend all the questions be mentioned? (Are there differences between children who attended all the sessions and those who only attended some? Even with this question we can't answer because we don't have that information)
Response: Thanks for the comment. The question is explicited in the first paragraph of the discussion section. It is possible that not in the sense that Reviewer 2 is suggesting, but the first limitation of the study was just not having all the responses of the initial sample. See L666-669.
References
C39 Review some incomplete references, such as: Thurstone, L.L.; Yela, M. CARAS-R. 2012.
Brown, L.; Sherbenou, R.J.; Johnsen, S.K.; Berna, G.·; Viena, ·; Oxford, ·; Boston, P.; Á msterdam, ·; Praga, ·; Copenhague, F.; et al. Grupo Editorial Hogrefe TONI-4. 2010
Vuarchex, T.; Yakovenko, P. Jungle Speed 1997.
Roubira, J.-L. Dixit; Libellud & Asmodee, 2008;
Faubet, J.; Hatesse, S. Mixmo; Asmodee, Sly Frog Games, 2009; - It is important to review all references in detail.
Response: Thanks for reviewing the references. We have changed all we have detected.
Good job.
I wish you all the best.
REFERENCES USED IN THE REVIEW
Baniqued, P. L., Kranz, M. B., Voss, M. W., Lee, H., Cosman, J. D., Severson, J., & Kramer, A. F. (2014). Cognitive training with casual video games: Points to consider. Frontiers in Psychology, 4(JAN), 1–19. doi:10.3389/fpsyg.2013.01010
Baniqued, P. L., Lee, H., Voss, M. W., Basak, C., Cosman, J. D., DeSouza, S., … Kramer, A. F. (2013). Selling points: What cognitive abilities are tapped by casual video games? Acta Psychologica, 142(1), 74–86. doi:10.1016/j.actpsy.2012.11.009
Benzing, V., Schmidt, M., Jäger, K., Egger, F., Conzelmann, A., & Roebers, C. M. (2019). A classroom intervention to improve executive functions in late primary school children: Too ‘old’ for improvements? British Journal of Educational Psychology, 89(2), 225–238. doi:10.1111/BJEP.12232
Dehn, M. J. (2017). How working memory enables fluid reasoning. Applied Neuropsychology: Child, 6(3), 245–247. doi:10.1080/21622965.2017.1317490
Estrada-Plana, V., Esquerda, M., Mangues, R., March-Llanes, J., & Moya-Higueras, J. (2019a). A Pilot Study of the Efficacy of a Cognitive Training Based on Board Games in Children with Attention-Deficit/Hyperactivity Disorder: A Randomized Controlled Trial. Games for Health Journal, 8(4), 265–274. doi:10.1089/g4h.2018.0051
Estrada-Plana, V., Esquerda, M., Mangues, R., March-Llanes, J., & Moya-Higueras, J. (2019b). A Pilot Study of the Efficacy of a Cognitive Training Based on Board Games in Children with Attention-Deficit/Hyperactivity Disorder: A Randomized Controlled Trial. Https://Home.Liebertpub.Com/G4h, 8(4), 265–274. doi:10.1089/G4H.2018.0051
Estrada-Plana, V., Montanera, R., Ibarz-Estruga, A., March-Llanes, J., Vita-Barrull, N., Guzmán, N., … Moya-Higueras, J. (2021). Cognitive training with modern board and card games in healthy older adults: two randomized controlled trials. International Journal of Geriatric Psychiatry, 36(6), 839–850. doi:10.1002/GPS.5484
Friedman, N. P., & Miyake, A. (2017). Unity and Diversity of Executive Functions: Individual Differences as a Window on Cognitive Structure.Cortex; a Journal Devoted to the Study of the Nervous System and Behavior, 86, 186. doi:10.1016/J.CORTEX.2016.04.023
Gashaj, V., Dapp, L. C., Trninic, D., & Roebers, C. M. (2021). The effect of video games, exergames and board games on executive functions in kindergarten and 2nd grade: An explorative longitudinal study. Trends in Neuroscience and Education, 25, 100162. doi:10.1016/j.tine.2021.100162
Klingberg, T. (2010). Training and plasticity of working memory. Trends in Cognitive Sciences, 14(7), 317–324. doi:10.1016/j.tics.2010.05.002
Kranz, M. B., Baniqued, P. L., Voss, M. W., Lee, H., & Kramer, A. F. (2017). Examining the roles of reasoning and working memory in predicting casual game performance across extended gameplay. Frontiers in Psychology, 8(MAR), 1–13. doi:10.3389/fpsyg.2017.00203
Loher, S., & Roebers, C. M. (2013). Executive Functions and Their Differential Contribution to Sustained Attention in 5- to 8-Year-Old Children. Journal of Educational and Developmental Psychology, 3(1), 51–63. doi:10.5539/jedp.v3n1p51
Miyake, A., Friedman, N. P., Emerson, M. J., Witzki, A. H., Howerter, A., & Wager, T. D. (2000). The Unity and Diversity of Executive Functions and Their Contributions to Complex “Frontal Lobe” Tasks: A Latent Variable Analysis. Cognitive Psychology, 41(1), 49–100. doi:10.1006/COGP.1999.0734
Spruijt, A. M., Dekker, M. C., Ziermans, T. B., & Swaab, H. (2018). Attentional control and executive functioning in school-aged children: Linking self-regulation and parenting strategies. Journal of Experimental Child Psychology, 166, 340–359. doi:10.1016/j.jecp.2017.09.004
Vita-Barrull, N., Guzmán, N., Estrada-Plana, V., March-Llanes, J., Mayoral, M., & Moya-Higueras, J. (2022). Impact on Executive Dysfunctions of Gamification and Nongamification in Playing Board Games in Children at Risk of Social Exclusion. Games for Health Journal, 11(1), 46–57. doi:10.1089/g4h.2021.0034
Vita-Barrull, N., March-Llanes, J., Guzmán, N., Estrada-Plana, V., Mayoral, M., Moya-Higueras, J., … Arias, S. (2022). The Cognitive Processes Behind Commercialized Board Games for Intervening in Mental Health and Education: A Committee of Experts. Games for Health Journal, 11(6), 414–424. doi:10.1089/g4h.2022.0109

Reviewer 3 Report
The introduction is well written and informative. The authors explained the research problem at the light of previous published studies, including pros and cons as previously found in the literature. I appreciate it, because it can be considered a good strategy to guide the reader and introduce the aims and hypotheses. However, despite the elegance, as I mentioned, the last paragraph of the introduction needs to be reformulated to introduce the aims. The last paragraph of the introduction needs to be written in a more clear way (line 75-77), eventually citing previous studies. According to me, this could be helpful to improve the global quality (high) of the study and the manuscript.
The methods section was extremely informative, well written and described. I appreciated the sampling. However, more info about “Evolution along sessions was evaluated with game session record sheets: an ad-hoc scale 167 was created for the present research” should be needed. I can suggest to add this scale in the supplementary materials. In this way, the study should be easily replicated in different countries with different populations. I advise to simplify the” procedure” paragraph adding a schematic figure. I also advise to check the English and carefully edit it (i.e. “persona” etc.).
I agree with the authors about the use of non-parametric test (U), but, reading the results, I found F-results. Moreover, the authors wrote about a mixed-model. The authors should clarify this part. Indeed, I suppose that they applied a mixed-model ANOVA. I suppose it, because I did not read about Lambda. I agree with the mixed model, but the authors need to specify it and assess the eventuality of the application of a MANOVA instead ANOVA.
Figure 3 is low quality, please improve it.
The discussion is very interesting and the authors added clinical implications and limitations.
Author Response
Response to Reviewer 3 Comments
C1. The introduction is well written and informative. The authors explained the research problem at the light of previous published studies, including pros and cons as previously found in the literature. I appreciate it, because it can be considered a good strategy to guide the reader and introduce the aims and hypotheses. However, despite the elegance, as I mentioned, the last paragraph of the introduction needs to be reformulated to introduce the aims. The last paragraph of the introduction needs to be written in a more clear way (line 75-77), eventually citing previous studies. According to me, this could be helpful to improve the global quality (high) of the study and the manuscript.
Response: Thanks for your words. We have added citations and we have added some sentences to clarify the objectives. See L91-93.
C2. The methods section was extremely informative, well written and described. I appreciated the sampling. However, more info about “Evolution along sessions was evaluated with game session record sheets: an ad-hoc scale 167 was created for the present research” should be needed. I can suggest to add this scale in the supplementary materials. In this way, the study should be easily replicated in different countries with different populations. I advise to simplify the” procedure” paragraph adding a schematic figure. I also advise to check the English and carefully edit it (i.e. “persona” etc.).
Response: Thanks for all the methods comments. As far as we have understood, other reviewers have in fact requested us to expand some parts of the methods section. So, we have decided to no summarize it. However, if Reviewer 3 stills think that the simplification of the procedure is needed, we are open to do it.
As Reviewer 3 has proposed, we have added the game session record sheets to the Appendinx. See L258 and Table S3.
C3. I agree with the authors about the use of non-parametric test (U), but, reading the results, I found F-results. Moreover, the authors wrote about a mixed-model. The authors should clarify this part. Indeed, I suppose that they applied a mixed-model ANOVA. I suppose it, because I did not read about Lambda. I agree with the mixed model, but the authors need to specify it and assess the eventuality of the application of a MANOVA instead ANOVA.
Response: Thanks for all the methods comments. It is possible that we have not explained this point deeply. We did not use a mixed-model ANOVA, but a linear mixed models. Linear mixed models are better than ANOVAs when it is supposed that we have multiple nested random effects (Hox, Moerbeek, & van de Schoot, 2017; Monsalves, Bangdiwala, Thabane, & Bangdiwala, 2020; Schielzeth et al., 2020). As different settings applied the methodology, we understood that we have the abovementioned situation. We have added the adequate information in the mansucript. See L 300-301.
C4. Figure 3 is low quality, please improve it.
Response: Thanks the comment. We have increased the quality of the image.
C5. The discussion is very interesting and the authors added clinical implications and limitations.
Response: Thanks for the kind comment.
REFERENCES USED IN THE REVIEW
Hox, J. J., Moerbeek, M., & van de Schoot, R. (2017). Multilevel Analysis. Third edition. | New York, NY : Routledge, 2017. |: Routledge. doi:10.4324/9781315650982
Monsalves, M. J., Bangdiwala, A. S., Thabane, A., & Bangdiwala, S. I. (2020). LEVEL ( Logical Explanations & Visualizations of Estimates in Linear mixed models ): recommendations for reporting multilevel data and analyses, 8, 1–9.
Schielzeth, H., Dingemanse, N. J., Nakagawa, S., Westneat, D. F., Allegue, H., Teplitsky, C., … Araya-Ajoy, Y. G. (2020). Robustness of linear mixed-effects models to violations of distributional assumptions. Methods in Ecology and Evolution, 11(9), 1141–1152. doi:10.1111/2041-210X.13434

Round 2
Reviewer 2 Report
Dears Authors
General comments
Congratulations. I think your work has been significant and the article is now more interesting.
Title
“Just play cognitive modern board and card games, it's going to be good for your executive functions: a randomized controlled trial with children at risk of social exclusion”
- I think it is more objective.
Introduction
C6 - Response: As far as we know, it is not so important how many weeks do last the intervention, but the number of sessions developed. As it is explained in P6 L227, we assured 12 gaming sessions. According to Klingberg (2010), a minium number of 10 sessions are needed to begin finding effects in cognitive training. In previous studies with specific populations, only 5 sessions were enough to find significant increases in group of children who played modern board games (Estrada-Plana, Esquerda, Mangues, March-Llanes, & Moya-Higueras, 2019a). Benzing et al., (2019) used the same temporality, showing significant results. In fact, with older adults, a game-based intervention within 6 weeks have found significant results (Estrada-Plana et al., 2021). In addition to this evidence, we thought that 6 weeks were easier to follow during the COVID-19 pandemic than a longer intervention. With more weeks to play, the more likely to have individual lockdowns and so on. Any way, we have added a sentence to justify this decission. See P6, L363-364. We have also added a paragraph in the limitations section (See L685-689).
- Lines 363-364 do not match the explanation. There must have been an error.
- As well as Lines 685-689:
- This question was not clear to me.
C11 - Response: Please, we kindly request Reviewer 2 to see our response to C6. However, we agree that a greater quantity of sessions should be advisible. So we have included it in the limitations section (see L666-669).
- The issue here is not about time, but about the number of sessions. I still think that individuals who integrated 3 and 4 weeks (6-8 hours) should not be included in the study. And even then I have doubts about the 10h. Because, as they say “According to Klingberg (2010), a minium number of 10 sessions are needed to begin finding effects in cognitive training. In previous studies with specific populations, only 5 sessions were enough to find significant increases in group of children who played modern board games (Estrada-Plana, Esquerda, Mangues, March-Llanes, & Moya-Higueras, 2019a).” A question that I would ask you to consider again.
C13 - Response: As the manuscript is long, we decided to sum up all the information requested by Reviewer 2. It can be found in P3 L149-159.
- It does not correspond again:
- I still consider a more detailed description of the sample to be essential.
C22- Response: Please, see response to Comment 6.
- The response to comment 6 does not answer my question.
C23- Lines 209-213- “We took into account three variables to select the games. The first one was the main cognitive process that the game is intended to activate when playing. This decision was taken by a subjective criteria by two researchers according to their experience on neuropsychology and on games. The experimental group played with games that, apparently, activated the basic executive functions.”- What subjective criteria? You should be clearer in this explanation. Not least because, in my opinion, this choice may have been responsible for your results. If it is not very well supported, the whole intervention can easily be equivocated.
- Your explanation did not fully answer my question. What were the subjective criteria? They should, in my view, be identified.
C31- “However, considering all the comments of Reviewer 2, we could specify in the discussion section which analysis we used to argue each idea.”
-It is, in my view, fundamental.
C38- Response: Thanks for the comment. The question is explicited in the first paragraph of the discussion section. It is possible that not in the sense that Reviewer 2 is suggesting, but the first limitation of the study was just not having all the responses of the initial sample. See L666-669.
- My question is another: Are there differences between children who attended all the sessions and those who only attended some? It would be interesting to analyze.
- I wish you a good job.
Author Response
Congratulations. I think your work has been significant and the article is now more interesting.
Response: Thank you so much. It has been a hard work. In fact, thanks to a comment of the Editor, we have accepted all the comments of Reviewer 2 in Round 1, in addition to the present comments in Round 2.
Title
“Just play cognitive modern board and card games, it's going to be good for your executive functions: a randomized controlled trial with children at risk of social exclusion”
- I think it is more objective.
Introduction
Response: We agree.
C6 - Response: As far as we know, it is not so important how many weeks do last the intervention, but the number of sessions developed. As it is explained in P6 L227, we assured 12 gaming sessions. According to Klingberg (2010), a minium number of 10 sessions are needed to begin finding effects in cognitive training. In previous studies with specific populations, only 5 sessions were enough to find significant increases in group of children who played modern board games (Estrada-Plana, Esquerda, Mangues, March-Llanes, & Moya-Higueras, 2019a). Benzing et al., (2019) used the same temporality, showing significant results. In fact, with older adults, a game-based intervention within 6 weeks have found significant results (Estrada-Plana et al., 2021). In addition to this evidence, we thought that 6 weeks were easier to follow during the COVID-19 pandemic than a longer intervention. With more weeks to play, the more likely to have individual lockdowns and so on. Any way, we have added a sentence to justify this decission. See P6, L363-364. We have also added a paragraph in the limitations section (See L685-689).
- Lines 363-364 do not match the explanation. There must have been an error.
- As well as Lines 685-689:
- This question was not clear to me.
Response: We apologize for the inconveniences. When we where modifying the manuscript, with such a great quantity of modifications, the word processor began to change the line's numbers from one time to another. So it is possible that some of our answers did not match the manuscript. In fact, according to the editor's comments, we have added more references to justify this decision. Please, see L238-241.
C11 - Response: Please, we kindly request Reviewer 2 to see our response to C6. However, we agree that a greater quantity of sessions should be advisible. So we have included it in the limitations section (see L666-669).
- The issue here is not about time, but about the number of sessions. I still think that individuals who integrated 3 and 4 weeks (6-8 hours) should not be included in the study. And even then I have doubts about the 10h. Because, as they say “According to Klingberg (2010), a minium number of 10 sessions are needed to begin finding effects in cognitive training. In previous studies with specific populations, only 5 sessions were enough to find significant increases in group of children who played modern board games (Estrada-Plana, Esquerda, Mangues, March-Llanes, & Moya-Higueras, 2019a).” A question that I would ask you to consider again.
Response: We apologize because of the great mistake we did. We are performing different studies like this one, and we exchanged the information about the number of weeks and sessions. As it is properly informed now in the manuscript (L109), the intervention lasted nine weeks, with two sessions per week equals to 18 session. We have also changed the information in the Procedure section (L238-239) and in the limitations accordingly (L516-517).
C13 - Response: As the manuscript is long, we decided to sum up all the information requested by Reviewer 2. It can be found in P3 L149-159.
- It does not correspond again:
- I still consider a more detailed description of the sample to be essential.
Response: Please, see L136-138. We have a full description of mean age and sex. All the sample met the fourth inclusion criteria, so all of them where at risk of social exclusion. We did not use any other indicator about the economic status or educative status of their parents because they met the criteria. But to enhance this section, we have added an attrition analysis comparing the retaind sample (67 participants) to the excluded sample (30) in the basic sociodemographics. See L139-145.
C22- Response: Please, see response to Comment 6.
- The response to comment 6 does not answer my question.
Response: As we said above, we did a mistake when we explained the number of sessions. We hope that the problem is fixed now.
C23- Lines 209-213- “We took into account three variables to select the games. The first one was the main cognitive process that the game is intended to activate when playing. This decision was taken by a subjective criteria by two researchers according to their experience on neuropsychology and on games. The experimental group played with games that, apparently, activated the basic executive functions.”- What subjective criteria? You should be clearer in this explanation. Not least because, in my opinion, this choice may have been responsible for your results. If it is not very well supported, the whole intervention can easily be equivocated.
- Your explanation did not fully answer my question. What were the subjective criteria? They should, in my view, be identified.
Response: We apologize for not being so clear in this question. We have added a sentence explaining it a bit (See L251) and a full table in the Appendix to justify the selection of the games (See Table S1).
C31- “However, considering all the comments of Reviewer 2, we could specify in the discussion section which analysis we used to argue each idea.”
-It is, in my view, fundamental.
Response: We have added in the discussion section the Table and Figures that justify any specific idea.
C38- Response: Thanks for the comment. The question is explicited in the first paragraph of the discussion section. It is possible that not in the sense that Reviewer 2 is suggesting, but the first limitation of the study was just not having all the responses of the initial sample. See L666-669.
- My question is another: Are there differences between children who attended all the sessions and those who only attended some? It would be interesting to analyze.
Response: This question is partially answered with the attrition analysis (See L139-145). In addition, we have compared the retained and the excluded sample in the study's outcomes. As it is explained in the manuscript (See L308-312), we only found 1 out of 12 significat differences. Thus, we interpret that no systematic sampling error have been made in the present research.
- I wish you a good job.
Response: Many thanks.

Reviewer 3 Report
The authors addressed all my concerns.
Author Response
Thanks for accepting the previous version of the manuscript.